# An In Vitro Study of the Healing Potential of Black Mulberry (*Morus nigra* L.) Extract in a Liposomal Formulation

Adriana Ramona Memete [1], Florina Miere (Groza) [2,*], Vasile Laslo [3], Cornelia Purcarea [4], Laura Vicas [2], Mariana Ganea [2], Angela Antonescu [2] and Simona Ioana Vicas [4,*]

1    Doctoral School of Biomedical Science, University of Oradea, 410087 Oradea, Romania
2    Faculty of Medicine and Pharmacy, University of Oradea, 10 P-ta 1 December Street, 410073 Oradea, Romania
3    Department of Environmental Engineering, Faculty of Environmental Protection, University of Oradea, 26 Gen. Magheru Street, 410048 Oradea, Romania
4    Department of Food Engineering, Faculty of Environmental Protection, University of Oradea, 26 Gen. Magheru Street, 410048 Oradea, Romania
*    Correspondence: florinamiere@uoradea.ro (F.M.); svicas@uoradea.ro (S.I.V.)

**Abstract:** Natural compounds are used in modern dermal treatments to avoid side effects commonly associated with conventional treatments. The aim of our study was to develop a liposomal formulation including black mulberry extract and to highlight its potential on the healing of normal human dermal fibroblasts (NHDF) in vitro using the scratch test. Mulberry-loaded liposomes (MnL) were prepared using a thin-film hydration method based on cholesterol (C) and phosphatidylcholine (PC) in a 1:3 ($w/w$) ratio. The liposomal formulation was characterized by analyzing its size, electric surface potential, morphology, entrapment efficiency, and in vitro healing effects. Also, the black mulberry fruits (*Morus nigra* L.) were characterized from point of view of polyphenolic compounds and antioxidant capacity by Ferric-Reducing Antioxidant Power (FRAP) assay. HPLC-DAD-MS (ESI$^+$) (high performance liquid chromatography-photodiode array detection-mass spectrometry (electrospray ionization)) analysis indicated the presence of phenolic compounds namely from hydroxybenzoic and hydroxycinnamic acids and flavonols. Among flavonols, quercetin-glucoside represented 50.56%, and chlorogenic acid was the predominant compound among hydroxycinnamic acids (37.06%). In vitro fibroblast wound closure was more effective with mulberry-loaded liposomes (MnL) than extracts of mulberries. According to our study, mulberry-loaded liposomes have been shown to be effective in wound healing and can be used as a natural treatment.

**Keywords:** *Morus nigra* L.; liposomes; wound healing; fibroblasts; scratch assay

## 1. Introduction

Mulberry fruits are considered very valuable products, due to their phytochemical composition. Black mulberry fruits are known to be rich sources of anthocyanins which, once isolated, could be used as a natural food coloring [1–3]. They are also known for their content of flavonoids, phenolic acids or vitamins [4]. Processing, unfortunately, can affect mulberry polyphenols and thus reduce their bio-accessibility [5,6].

Phenolic acids are known to exhibit significant biological activity, such as antioxidant potential, anti-inflammatory, anti-dyslipidemic, antimicrobial or vascular protective effect [7]. Therefore, the current interest in the bioactive phenolic compounds present in black mulberry fruits has increased the attention in terms of optimizing the extraction procedures of bioactive phytochemicals [1,3,5,8]. Several studies on the phytochemical composition of black mulberry (*M. nigra*) fruits have led to the development of natural antioxidant formulations intended for food supplements, food, cosmetics or with different applications [6,9].

In addition to anthocyanins, flavonoids and phenolic acids, vitamin C was also identified in *M nigra*, which is a strong antioxidant with an important role on the skin [10–12].

The application of vitamin C to the skin is limited by its hydrophilic property and the hydrophobic skin layer based on an organized lipid matrix. Nanocarriers such as liposomes offer advantages for topical and transdermal application of encapsulated hydrophilic molecules by increasing their solubility and diffusion in the stratum corneum, as well as increasing their stability [13].

Fibroblasts are the most abundant cells located in the dermal layer, along with other cells of the dermis, such as: fibrocytes, myofibroblasts [14,15]. A fibroblasts main characteristic is its ability to synthesize and secrete collagen, in addition to remodeling the extracellular matrix and communicating with other cells [16,17]. Remodeling of extracellular matrix is supported by the synthesis of cleaved metalloproteinases and their inhibitors [16].

The in vitro evaluation of different extracts or compounds on the wound healing effect, known as the scratch assay, is an easy and inexpensive method that provides valuable information about the biological properties of the tested sample [18,19]. Also, currently, a major interest consist in the formulation of transport vectors such as lipid vesicles, liposomes or nanoliposomes [20,21]. The major advantage of liposomal formulas is that they are biocompatible with the human body, for example the use of phospholipids shows major compatibility with the topical applications of these formulations [20,22,23]. Another advantage of using these transport systems is that at the level of the dermis they can release the entrapped compounds in a controlled and continuous manner, which allows obtaining a reproducible treatment [24,25]. In addition, the formulation of liposomes with included extracts allows the protection of biologically active compounds against environmental factors (temperature, humidity, light, etc.) or those related to the human body (pH, enzymes), increasing the effectiveness of the treatment [24]. As a transport system for skin treatment, liposomes have also been shown to have some disadvantages, including their low stability over time, in some instances, their low efficiency of encapsulation, and the need for methods that can be performed easily and reproducibly [26]. In order to rectify these disadvantages, the formulation technique should be adapted according to the compounds to be included in the liposomes [26].

To our knowledge, the black mulberry extract has never been studied for its in vitro wound healing capacity, both as an extract rich in polyphenolic compounds and encapsulated in liposomes. In this context, the first purpose of our study was to characterize the mulberry extract in terms of bioactive compounds (phenolic acids, flavonoids, ascorbic acid) and antioxidant capacity. In vitro testing of the proliferation and healing effect of both forms, the black mulberry extract (MnE) and encapsulated in a liposome (MnL) using the scratch method on normal human dermal fibroblasts (NHDF) was also pursued. The steps of experimental design of our study regarding to the characterization of phytochemical profile of mulberry extract (MnE) and its inclusion in liposomal formulas and in vitro testing of the wound healing is presented in Figure 1.

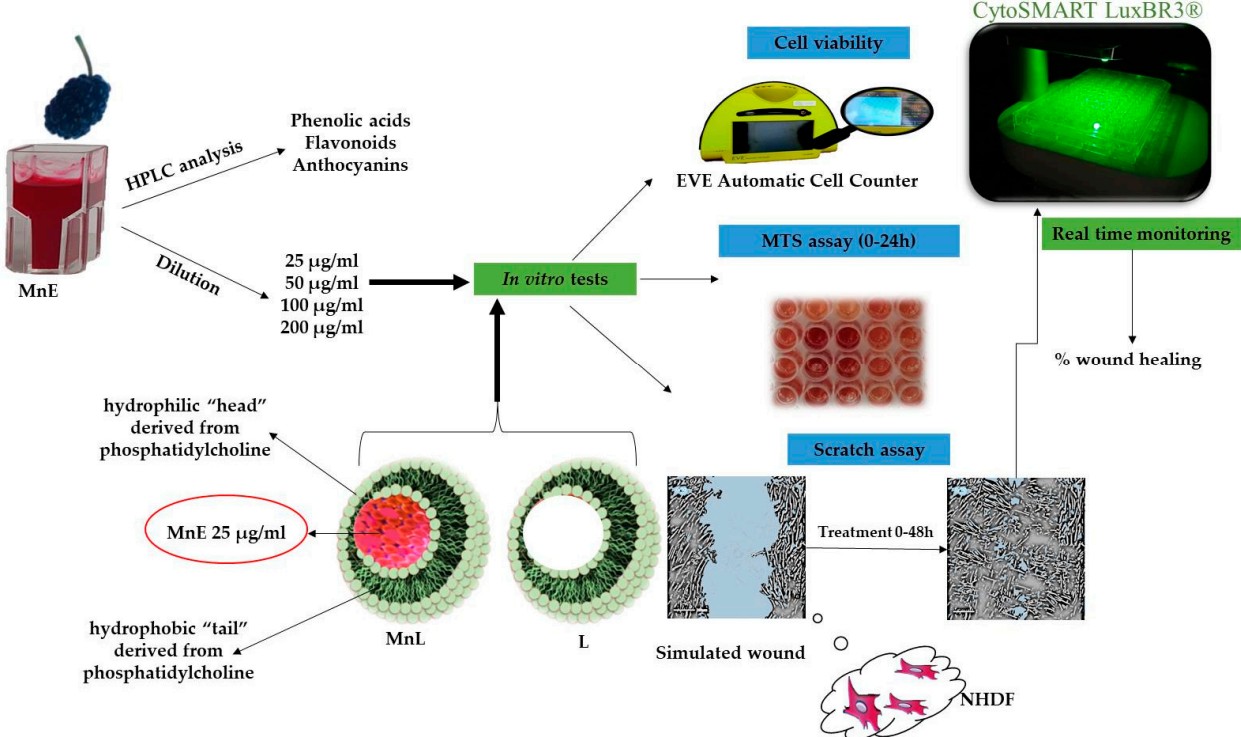

**Figure 1.** Experimental design. MnE—*M. nigra* L. extract; MnL—liposomes loaded with *M. nigra* L. extract (25 µg/mL), L—empty liposome.

## 2. Materials and Methods

### 2.1. Chemicals

The HPLC reference standards of gallic acid, chlorogenic acid, rutin, cyanidin and solvents (acetonitrile, acetic acid) were purchased from Sigma Aldrich (St. Louis, MO, USA). The water for the HPLC analysis was purified using a Milli-Q system (Merck Millipore). For spectrophotometric methods, Ascorbic acid, Quercetin, 6-hydroxy-2,5,7,8-tetramethylchroman-2-carboxylic acid, Folin & Ciocalteu's phenol-Aldrich were purchased from Sigma Aldrich (St. Louis, MO, USA) and 2,4,6-Tris(2-pyridyl)-s-triazine, aluminium chloride were purchased from Fluka (Charlotte, NC, USA). Cholesterol and Phosphatidyl-choline were obtained from Sigma Aldrich (St. Louis, MO, USA), dichloromethane and methanol were of analytical grade.

### 2.2. Preparation of Mulberry Extract

Fresh mulberry fruits (5 g) were extracted with 5 mL acidified methanol (trifluoroacetic acid (TFA), 0.1%) by homogenization (Heidolph homogenizer, Silent Crusher M) at 12,000 rpm for one minute. The homogenized samples were then centrifuged (NÜVE NF 200 BENCH TOP CENTRIFUGE, Turkey) at 5000 rpm for 20 min. The pellet was subjected to repeated extraction until complete decolorization of the solvent. The collected supernatant was concentrated, by evaporation of methanol, in the vacuum rotary evaporator (Heidolph Rotary Evaporator, Laborota 4000 rotavapor, Schwabach, Germany) at 40 °C, according to Bunea et al. (2011) [27] with minor modifications [6]. The *Morus nigra* extract (MnE) was freeze-drying using Christ Alpha 1-2 LDplus freeze dryer (Osterode am Harz, Germany) and used for the cell culture analysis.

### 2.3. HPLC-DAD-MS (ESI⁺) Characterization of the Extract from the Point of View of Anthocyanin Content

For the separation and identification of phenols by HPLC, 0.5 g of mulberries were extracted with acidified methanol with 1% HCl of 37% by vortexing for 1 min with Heidolph Reax top vortex (Heidolph Instruments GmbH & Co., Schwabach, Germany), sonicated

for 15 min in the Elmasonic E 15 H sonication bath, and centrifuged at 10,000 rpm, for 10 min and 40 °C in the Eppendorf AG 5804 centrifuge, SUA. The supernatant was filtered through filter paper and kept in the dark. The pellet was subjected to repeated extraction, under the conditions specified above, until complete discoloration. The cumulative extract was concentrated, by evaporation at low pressure on the Heidolph rotary evaporator, to a volume of 2 mL, it was filtered through a 0.45 μm Chromafil Xtra nylon filter and 20 μL were injected into the HPLC system [27].

Identification of the compounds in the black mulberry fruit extract was performed using an HP-1200 liquid chromatograph equipped with a quaternary pump, autosampler, DAD detector, and single-quadrupole API electrospray MS-6110 detector (Agilent-Technologies, Santa Clara, CA, USA). For the detection of phenolic compounds, full scan ESI positive ionization mode was used in the following working conditions: capillary voltage 3000 V, temperature 3500 C, nitrogen flow 7 L/min and $m/z$ 120–1200. The column was a Kinetex XB-C18 (5 μm; 4.6 × 150 mm i.d.) from Phenomenex, Torrance, CA, USA. The mobile phase A was water acidified by 0.1% acetic acid and mobile phase B was acetonitrile acidified with 0.1% acetic acid. The following multistep linear gradient was applied: 0 min, 5% B; 0–2 min, 5% B; 2–18 min, 5–40% B; 18–20 min, 40–90% B; 20–24 min, 90% B; 24–25 min, 90–5% B; 25–30 min, 5% B. Total analysis time was 30 min, flow rate 0.5 mL/min and oven temperature 25 ± 0.5 °C.

Mass spectrometric detection of positively charged ions was performed using scanning mode. The applied experimental conditions were: gas temperature 350 °C, nitrogen flow 7 L/min, capillary voltage 3000 V and $m/z$ 120–1200. Chromatograms were recorded at wavelength λ = 280, 340 and 520 nm, and data acquisition and interpretation of results was done using Agilent ChemStation software (B.02.01SR2, Santa Clara, CA, USA).

Phenolic compounds were identified by comparing retention time, UV-Vis absorption and mass spectra with those of standard compounds and literature data [1,28]. Based on the spectral characteristics of phenolic compounds, the wavelength λ = 280 nm is specific for some phenolic acids, flavanol monomers and polymers, while λ = 340 nm for hydroxycinnamic acids and flavonols, and λ = 520 nm for anthocyanins.

For the quantification of phenolic compounds, calibration curves were made by injecting five different concentrations of standard substances (ranging from 1 to 100 μg/mL) dissolved in methanol: Gallic acid ($R^2$ = 0.9978), LOD (limit of detection) = 0.35 μg/mL, LOQ (limit of quantification) = 1.05 μg/mL; Chlorogenic acid ($R^2$ = 0.9937), LOD = 0.41 μg/mL, LOQ = 1.64 μg/mL; Rutin ($R^2$ = 0.9981), LOD = 0.21 μg/mL, LOQ = 0.84 μg/mL and Cyanidine ($R^2$ = 0.9951), LOD = 0.36 μg/mL, LOQ = 1.44 μg/mL [29]. The peak area of each component in the mulberry extract was acquired from chromatogram and the abundance of each compound was calculated from its corresponding calibration curve.

### 2.4. Determination of Total Phenol Content (TPc)

The total phenolic content of MnE was determined by the Folin-Ciocalteu method with some modifications [30,31]. Briefly, MnE (100 μL) was incubated with 1700 μL of distilled water, 200 μL of Folin-Ciocalteu reagent (freshly prepared, 1:10 dilution, $v/v$) and 1000 μL of 7.5% $Na_2CO_3$ solution for 2 h in the dark at room temperature. Absorbance was measured at 765 nm (Shimatzu mini-UV-Vis spectrophotometer) and results are expressed as milligrams of gallic acid equivalents (GAE)/100 g of dry weight (dw) using gallic acid as standard.

### 2.5. Determination of Total Flavonoids Content (TFc)

The total flavonoid content was determined by the colorimetric method of aluminum chloride [6]. In a 10 mL volumetric flask, the MnE (1 mL) was mixed with 4 mL of distilled water and 0.3 mL of 5% $NaNO_2$, and allowed to react for 5 min. Then, 0.3 mL of 10% $AlCl_3$ was added, and after 6 min, 2 mL of 1 M NaOH was added to the mixture, and the content of flask was completed with distilled water to obtain a final volume of 10 mL and thoroughly mixed. The entire composition in the flask was homogenized and allowed to

react together for an additional 15 min, and then, the absorbance at a wavelength of 510 nm was determined. The results were expressed in mg QE (quercetin equivalents)/100 g dw.

## 2.6. Ascorbic Acid Determination

The ascorbic acid content of mulberry fruits was determined using the spectrophotometric method with xylene extraction [32]. Five grams of fresh mulberry fruit was homogenized in 3% metaphosphoric acid solution, followed by filtering. After transferring 2 mL of the obtained filtrate into a separating funnel, 2 mL of acetate buffer pH = 4 was added. Then 3 mL of 2, 6 dichlorophenol indophenol solution (0.0007 M) and 10 mL of xylene were added and vigorously shaken for 30 s. When the phases have separated, the absorbance of the organic phase was recorded at a wavelength of 520 nm, against xylene as a control. The results are quantified using the calibration curve, made with ascorbic acid of different concentrations (0; 0.5; 0.75; 1.0; 1.5; 2 mg/mL) and are expressed in mg ascorbic acid/100 g fw.

## 2.7. Determination of Antioxidant Capacity
FRAP Assay

The FRAP assay tested the antioxidant power of samples, based on the ability of the extract to reduce $Fe^{3+}$ from the tripyridyltriazine complex $Fe(TPTZ)^{3+}$ to the blue color complex-$Fe(TPTZ)^{2+}$ in acidic medium, with maximum absorption at 595 nm, method described by Benzie and Strain, 1996 with minor modifications, according to Memete et al. (2022) [6,33]. Briefly, the mulberry extract (100 μL) was allowed to react with 500 μL FRAP reagent and 2 mL distilled water for 1 h in the dark. The FRAP reagent was prepared using 300 mM acetate buffer (pH 3.6), 20 mM $FeCl_3$ and 10 mM TPTZ reagent in a ratio of 10:1:1 (*v/v/v*). A calibration curve was made with different concentrations of Trolox and the results were expressed in mg Trolox equivalent (TE)/100 g dw.

## 2.8. Formulation of Liposomes with Mulberry Extract and Their Characterization
2.8.1. Formulation of Liposomes Using the Lipid Film Hydration Method

Liposomes were formulated using the lipid film hydration method. The lipid part of the liposomes is composed of cholesterol (C) and phosphatidylcholine (PC) in a ratio of 1:3 (*w/w*). The hydrophilic phase is represented by the phosphate buffer pH = 7.6 which was used to hydrate the lipid film [34,35]. Briefly, C and PC were dissolved and mixed in an organic solvent (20 mL) consisting of dichloromethane and methanol (3:2 *v/v* ratio) to obtain a homogenous mixture of lipids. Then, the solvent was removed using the Heidolph Rotary Evaporator (Laborota 4000 rotavapor, Schwabach, Germany) maintaining the parameters of 37 °C and 100 rpm to yield a lipid film. Hydratation of the dry lipid film was achieved with phosphate buffer pH = 7.6 (10 mL) followed by sonication for 30 min, centrifugation for 3 min at 4050 rpm. The supernatant is collected and then centrifuged again for 30 min at 10,000 rpm [24].

Two types of liposomes were formulated, one without included mulberry extract (L) and another including 25 μg/mL of mulberry extract (MnL).

2.8.2. Microscopic Characterization of the Formed Liposomes

The formulation of liposomes was confirmed, by visualized them under a microscope according to the method described by Miere (Groza) et al., 2021 [24]. Thus, an Olympus CX40 (Tokyo, Japan) inverted light microscope was used, through a 40X objective in phase-contrast mode, and the images were captured by a Hitachi CCD camera.

2.8.3. Characterization of the Size and the Electric Surface Potential of the Liposomes by DLS

The dynamic light scattering (DLS) method was applied to determine the diameter, distribution and Zeta potential of formulated liposomes using a Zetasizer Nano ZS (Malvern Instruments, Worcestershire, UK). Polystyrene cells with an optical path of 1 cm were used

for diameter measurements. Disposable folded capillary cells were used to determine the electrical surface charge (Zeta potential). All measurements were made in triplicate for both L and MnL.

### 2.8.4. Determination of the Entrapment Efficiency (EE%) of the Mulberry Extract in Liposomes

The EE% determination was carried out according to the method proposed by Miere et al., 2021 and Gibis et al., 2016 [24,36]. Thus, the EE % was calculated using Formula (1) which is based on the determination of the total phenols content of the mulberry extract before and after its inclusion in liposomes. The total polyphenols are determined by the Folin Ciocalteu method, and the results are expressed in milligrams of gallic acid equivalent (mg GAE/mL).

$$EE\% = \frac{TPc\_MnL}{TPc\_MnE} \times 100 \tag{1}$$

where, TPc_MnL means total phenols content from mulberry extract entrapped in liposomal forms and TPc_MnE means total phenols content of the mulberry extract before entrapment in liposomes formulation.

### 2.9. Cell Culture and Treatment

In vitro, mulberry extracts (MnEs) at different concentrations (25, 50, 100 and 200 pg/mL), as well as liposomes containing mulberry extracts (MnL), were evaluated on normal human dermal fibroblasts (NHDF) migration using the scratch method. Normal human dermal fibroblasts and the specific culture kit (Fibroblast Growth Medium-2 BulletKit) were purchased from Lonza Pharma & Biotech (Basel, Switzerland). The culture medium was Dulbecco's Modified Eagle Medium containing 10% FBS (fetal bovine serum), gentamicin 50 mg/mL, amphotericin 50 mg/mL and recombinant fibroblast growth factor hFG (CC-4065) 1 mg/mL.

Treatments were applied both with mulberry extracts (MnEs) with concentrations between 25–200 μg/mL and with the MnL sample that includes mulberry extract with a concentration of 25 μg/mL. A CTRL sample (fibroblasts without treatment), liposomal sample (L) (fibroblasts treated only with empty liposomes) and a positive control (fibroblasts treated with 50 μg/mL Allantoin, ALA_50) were considered for this type of treatment.

Using the CytoSMART Lux3BR® device, all the samples were monitored in real time and the percentages of healing of the simulated wound were calculated.

### 2.9.1. Cell Viability Assay

The cell viability was performed using trypan blue exclusion assay [37]. NHDF cells were seeded into 24-well plates at $1 \times 10^4$ cells/well and maintained at 37 °C with 5% $CO_2$ for 24 h and then the cells were treated with MnE extract at final concentration of 25, 50, 100 and 200 μg/mL, liposomes with and without mulberry extract for 24 h. The cells were trypsinized (Trypsin/EDTA solution (0.25 mg/mL), LONZA), neutralized using TNS—Trypsin Neutralizing Solution, (LONZA) and centrifuged (1000 rpm/5 min). The pellet was suspended in cell medium and the viability of cells was determined using an EVE Automatic cell counter (NanoEnTek Inc., Seoul, Republic of Korea). The cell viability was calculated according to Formula (2) and the results were expressed as percentage of cell viability of treated cells against control [19]. The measurements were performed in triplicate, and the data are represented as mean ± standard deviation (SD).

$$Cell\ viability\ (\%) = \frac{number\ of\ living\ cells}{number\ of\ total\ cells} \times 100 \tag{2}$$

### 2.9.2. Cell Proliferation Assay (MTS)

The MTS assay is based on the reduction of the tetrazolium compound MTS by viable cells (human fibroblasts) to generate a colored formazan dye that is soluble in the cell culture medium.



To perform the cell proliferation assay, cells were seeded in sterile 98-well plates at a density of $2 \times 10^5$/mL by adding 50 µL of suspension cell growth medium. After 24 h, the samples (MnEs, MnL, L, ALA_50 and CTRL) were applied in a volume of 50 µL.

The proliferation was determined at 24 h. Briefly, 10 µL of MTS reagent containing (3-(4,5-dimethylthiazol-2-yl)-5-(3-carboxy-methoxyphenyl)-2-(4-sulfophenyl) 2H tetrazolium) and PMS (phenazine) in a ratio of 20:1 was added. After 2 h, the formazan dye was quantified by measuring the absorbance at 492 nm using a microplate reader (Stat Fax 2100, Palm City, USA), while the reference wavelength was 630 nm [38,39]. Results were expressed as percentage of cell viability compared to control. Tests were performed in triplicate.

2.9.3. In Vitro Testing of the Healing Effect of Mulberry Extracts and Its Liposomes Formulated Using the Scratch Method

The scratch method was used to investigate the wound healing potential of both black mulberry extract (MnE) at different concentrations (25 g/mL, 50 g/mL, 100 g/mL, 200 g/mL) and liposomes loaded with mulberry extract (MnL)For this, 24-well plates were seeded at a cell density of $4 \times 10^5$/cm$^2$. After 48 h, when the cells reached confluence, scratching was done vertically in each well with a sterile 50 µL pipette tip. The sloughed cells was removed by washing with 250 µL of HEPES buffer and subsequently the samples were applied in wells plate in triplicate. Then, the well plate was placed on the CytoSMART Lux3BR® device inside a culture incubator (37 °C and 5% CO$_2$).

The wound healing treatments was monitored at different time points: T0h, T12h, T24h, T36h and T48h. As quantitative parameters, wound closure by area (Wd_A) (%) were analyzed [19,40]. The evaluation of wound closure per area (%), was calculated for each sample, at each time, applying Formula (3):

$$\text{Wd\_A (\%)} = \frac{\text{Average area of wound\_sample (µm}^2\text{) at time t}}{\text{Average area of wound\_sample (µm}^2\text{) at time t} = \text{T0}} \times 100 \qquad (3)$$

*2.10. Statistical Analyses*

The data of analysis are represented as mean value ± SD (standard deviation) and all experiments were performed in triplicate (n = 3). The data were subjected to analysis by one-way ANOVA (Tukey's multiple comparison test) at $p < 0.05$ significant level. The images were uploaded to the CytoSMART cloud and the scratch area was subsequently used to quantitatively assess the wound healing process over time.

**3. Results and Discussions**

*3.1. Phytochemical Characterization of the MnE*

The phenols compounds identified from black mulberry fruits are divided into hydroxybenzoic acids (compound **1**, **2**), five hydroxycinnamic acids (compounds **7**–**9** and **11**, **12**), one phenolic acid glucoside (compound **10**), five flavonoid glycosides (compounds **6**, and **13**–**16**) and two anthocyanins (compound **4** and **5**). Quantitative data on the composition of phenolic compounds expressed in µg/g fw from the black mulberry extract and the elution order are presented in Table 1, and the HPLC chromatograms are shown in Figure S1.

In mulberry fruits, glycosylated flavonols were the predominant compounds (54.64%), while hydroxycinnamic acids predominate over hydroxybenzoic acids of 23.62% and 14.05%, respectively. Compounds **7** and **8** have UV spectra at λmax 322 nm and their MS spectra exhibited [M+H]$^+$ at $m/z$ 355, indicating isomers of chlorogenic acid. Similar results were obtained for Caffeoylquinic acid isomers identified in *Ilex paraguariensis* leaves and stemlets [41]. Chlorogenic acid (compound **7**) was the predominant compound from phenolic acids class found in mulberry fruits in the amount of 37.06%.

Quinic acid can form esters with ferulic acid, obtaining different isomers known as Feruloylquinic acid. Three Feruloylquinic acid derivatives (compound **9**, **11**, **12**) showed

the [M+H]$^+$ at *m/z* 369, comparable with other studies [42]. Compound **10** was identified as vanillic acid glucoside, due to molecular ion at *m/z* = 331.

**Table 1.** HPLC-DAD-MS (ESI$^+$) tentative phenolic compounds identified and quantified (µg/g fw) in the mulberry extract (MnE).

| Peak No. | $R_t$ (min) | UV $\lambda_{max}$ (nm) | [M+H]$^+$ (m/z) | Phenolic Compounds | Subclass | MnE (µg/g fw) * |
|---|---|---|---|---|---|---|
| 1. | 3.05 | 270 | 155 | Dihydroxybenzoic acid | Hydroxybenzoic acid | 1143.714 ± 115.38 |
| 2. | 3.84 | 270 | 139 | Hydroxybenzoic acid | Hydroxybenzoic acid | 177.988 ± 16.99 |
| 3. | 9.36 | 280 | 579 | Procyanidin dimer | Flavanol | 672.943 ± 67.94 |
| 4. | 10.91 | 520 | 449 | Cyanidin 3-*O*-glucoside | Anthocyanins | 947.464 ± 93.96 |
| 5. | | | 595 | Cyanidin 3-*O*-rutinoside | Anthocyanins | |
| 6. | 12.02 | 360, 255 | 773, 303 | Quercetin 3-glucosyl-(1->2)-rhamnoside-7-glucoside | Flavonol | 204.503 ± 19.59 |
| 7. | 12.35 | 322 | 355 | 5-Caffeoylquinic acid (chlorogenic acid) | Hydroxycinnamic acid | 1078.456 ± 109.08 |
| 8. | 12.65 | 322 | 355 | 1-Caffeoylquinic acid | Hydroxycinnamic acid | 587.256 ± 57.76 |
| 9. | 13.19 | 323 | 369 | 3-Feruloylquinic acid | Hydroxycinnamic acid | 202.347 ± 19.93 |
| 10. | 13.73 | 270 | 331 | Vanillic acid-glucoside | Phenolic glucoside | 408.647 ± 40.89 |
| 11. | 14.50 | 323 | 369 | 4-Feruloylquinic acid | Hydroxycinnamic acid | 609.269 ± 60.93 |
| 12. | 15.17 | 323 | 369 | 5-Feruloylquinic acid | Hydroxycinnamic acid | 431.909 ± 43.71 |
| 13. | 15.44 | 360, 255 | 611, 303 | Quercetin 3-*O*-rutinoside (rutin) | Flavonol | 1654.227 ± 166.02 |
| 14. | 16.11 | 360, 255 | 465, 303 | Quercetin 3-*O*-glucoside | Flavonol | 3402.965 ± 339.97 |
| 15. | 17.10 | 360, 255 | 449, 303 | Quercetin 3-*O*-rhamnoside | Flavonol | 459.747 ± 46.07 |
| 16. | 18.16 | 350, 250 | 473, 287 | Kaempferol 3-O-glucuronide | Flavonol | 335.817 ± 32.98 |

\* The values are presented as mean ± SD, (n = 3). $R_t$—Retention time; MnE—Morus nigra extract.

Quercetin in conjugated forms linked to sugars were detected: Quercetin 3-glucosyl-(1->2)-rhamnoside-7-glucoside (**6**), Quercetin 3-*O*-rutinoside (**13**), Quercetin 3-*O*-glucoside (**14**), Quercetin 3-*O*-rhamnoside (**15**). The UV-vis spectrum shown maximal absorption (λmax) of typical of flavonol derivative at 360 and 255 nm [29,43]. All quercetin derivatives shown one fragment ion *m/z* 303 characteristic to quercetin aglycon, and the fragments ions *m/z* corresponding to the sugar moiety. Among flavonoids, quercetin 3-*O*-glucoside is predominant, followed by Rutin (Table 1), results in agreement with the data published by Pawlowska et al., 2008 [8]. Kaempferol 3-*O* –glucuronide showed the [M+H]$^+$ at *m/z* 473 and the main fragment ion at *m/z* 287 indicated an aglycon moiety that was produce by losing a 186 amu, corresponding tu glucosiduronic acid. The UV-vis spectra of kaempferol 3-*O*-glucuronide shown λmax at 350 and 250 nm. 3-*O*-Glucuronidation of kaempferol caused hypsochromic Band I λmax shift of 17.6 nm [44,45]. In contrast to glycosylated quercetin derivatives, Kaempferol 3-*O*-glucuronide was in low amounts in our study. Another study demonstrated that the main compounds present in *M. nigra* fruits come in the different glycosylated forms of Quercetin and Kaempferol, respectively, and the content of quercetin-glycoside was found to be higher than the amount of Kaempferol-*O*-rutinoside [46]. Two anthocyanins were identified in mulberry fruits, namely Cyanidin-3-*O*-glucoside and Cyanidin-3-*O*-rutinoside. Our results are in agreement with Kim and Lee (2020) study, which showed that Cyanidin glucoside and Cyanidin rutinoside are major anthocyanins present in mulberry fruits, although they also identified the presence of other anthocyanins in twelve mulberry fruit cultivars from Korea [28].

The major profile of phenolic compounds identified in *M. nigra* fruits, depending on the amount recorded, was given by phenolic acids, followed by flavonols and only then, anthocyanins [1,46]. In other studies, a different pattern resulted, where anthocyanins were the compounds with the highest amount compared to the amount of flavonols [3,4].

Due to the increased interest in the systemic and topical applications of phenolic compounds extracted from plants, with a protective effect on the skin, numerous studies in the literature have proven that the phenolic acids, flavonoids and anthocyanins present

in black mulberry fruits have an effective protective effect in reducing the oxidative stress of tyrosinase and inflammation, contributing to the repair of dermal tissue in case of damage, stimulating cell proliferation, or can act as protective agents against UV rays and natural brightening [1,7,47–49]. Likewise, topical application with chlorogenic acid or quercetin inhibits erythema formation [50,51], quercetin treatments have the ability to attenuate cellular senescence of dermal fibroblasts [52], while cyanidin-3-*O*-glucoside also has a protective effect on the skin, fighting against UVB radiation [53]. Thus, the phenolic compounds present in black mulberry fruits, with a strong antioxidant, anti-inflammatory and protective effect on the skin, can provide a basis for the development of a powerful natural therapy to reduce skin disorders induced by oxidative stress.

### 3.2. Determination of Total Phenols, Flavonoids, Ascorbic Acid Content and Antioxidant Capacity

The amount of total phenols and flavonoids, ascorbic acid and antioxidant capacity, of *M. nigra* fruits, depends on a variety of factors such as the extraction method, the fruit cultivar, the stage of maturity of the fruit, as well as the climatic conditions [1,10,11,54,55].

The content of total phenols, total flavonoids, ascorbic acid and antioxidant capacity determined by the FRAP method of MnE are presented in Table 2.

**Table 2.** Total phenols, flavonoids, ascorbic acid content and antioxidant capacity determined by the FRAP method of MnE *.

| Samples | TPc (mg GAE/100 g dw) | TFc (mg QE/100 g dw) | Ascorbic acid (mg AA/100 g fw) | FRAP (mmol TE/100 g dw) |
|---|---|---|---|---|
| MnE | 1460.37 ± 61.42 | 296.48 ± 79.53 | 43.71 ± 1.481 | 80.23 ± 0.56 |

* The results are presented as mean ± SD (n = 3). TPc—Total phenols content; Gallic acid equivalent (GAE); TFc—Total flavonoids content; QE—Quercetin; AA—Ascorbic acid; FRAP—Ferric-Reducing Antioxidant Power; TE—Trolox equivalent; MnE—*Morus nigra* extract.

The results obtained in this study are similar to those reported in the literature [10,11]. *M. nigra* has the highest total phenolic content (368.2 mgGAE/100 g FW) and total flavonoid content (247.9 mg CTE/100 g FW), when compared to fruits of *M. alba* [56].

In the study of Ercisli and Orhan, 2007, the content of total polyphenols, total flavonoids, as well as ascorbic acid, were quantified from the fruits of *M. nigra*, *rubra* and *alba*, and the highest value of TPc and TFc was recorded for *M. nigra* (1422 mg GAE/100 g dw and 276 mg QE/100 g fw, respectively), followed by *M. rubra* and *M. alba*. The highest value of ascorbic acid, was recorded in *M. alba* (22.4 mg/100 mL), followed by *M. nigra* and *M. rubra* with 21.8 and 19.4 mg/100 mL respectively [11]. Also, Jiang and Nie, 2015 determined the content of ascorbic acid in the three types of mulberry fruits, grown in China where *M nigra* had the highest content of (48.4 mg/100 g fw), compared to the fruits of *M. rubra* and *alba* (5.64 mg/100 g fw and 6.01 mg/100 g fw, respectively) [10]. Ascorbic acid a powerful water-soluble antioxidant compound has the ability to neutralize and scavenge the reactive oxygen species and it has been shown that has an important role in protecting the skin against oxidative damage such as pollutants and UV radiation [10,57,58].

In a study by Kim and Lee, 2020, the antioxidant capacity of 12 mulberry genotypes by the FRAP method revealed values ranging from 1.33 to 82.87 mg TE/g dw [28].

### 3.3. Formulation of Liposomes with Mulberry Extract and Their Characterization

3.3.1. Formulation of Liposomes Using the Lipid Film Hydration Method and Microscopic Characterization of the Formed Liposomes

In addition to identifying the optimal concentration of mulberry extract with healing effect (MnE 25 g/mL), the main aim was to incorporate this extract into liposomal formulas, characterize them, and evaluate the healing potential of it. To confirm that the lipid vesicles were formed spontaneously following the hydration of the lipid film according to the authors Miere (Groza) et al., 2021 [24] and Popovska et al., 2020 [59], microscopic analysis was used.

Thus, following the microscopic analysis, it can be seen in Figure 2 that the empty liposomes (L) and mulberry-loaded liposomes (MnL) have a spherical or round appearance, and are not agglomerated, repelling each other. Our results are according with those of other authors [24,59] who obtained liposomes with the similar characteristics.

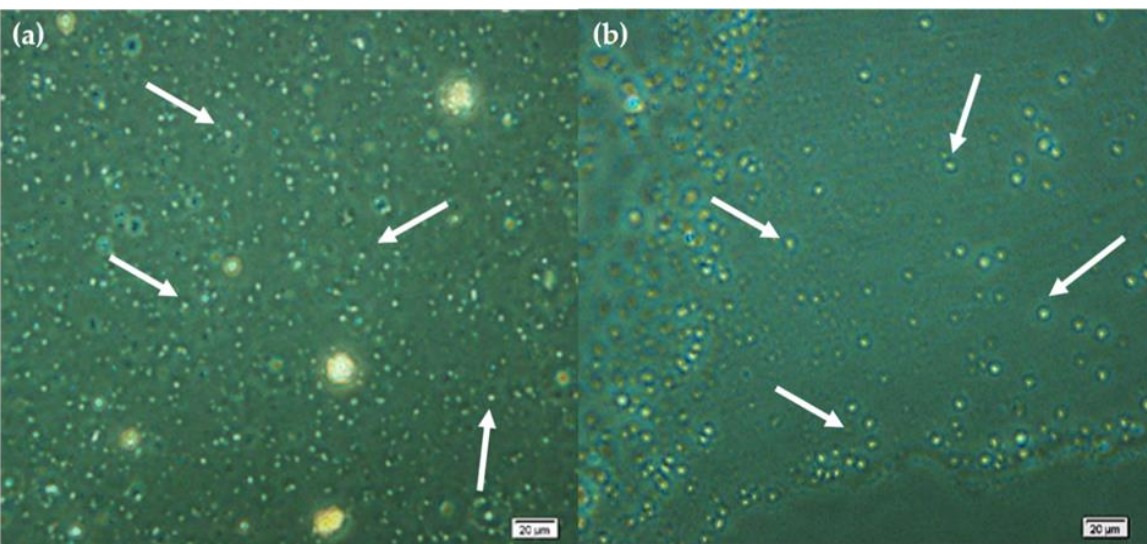

**Figure 2.** Microscopic images obtained with an Olympus CX40 inverted light microscope, through a 40× objective in phase contrast mode, captured by a Hitachi CCD camera. (**a**) L—empty liposomes, (**b**) MnL—mulberry-loaded liposomes. The white arrows highlight in both cases the spherical, round, dispersed liposomes.

Mohammadabadi et al., 2018 [20], report that phospholipids and sterols such as cholesterol are the most common ingredients used in liposome formulations, and that hydrophilic phase buffer systems are most commonly used. The ratio of phosphatidylcholine to cholesterol of 3:1 *w/w* that was used in the formulation of our liposomes (MnL) is one that confers stability to the formulated vesicles according to Asprea et al., 2019 [60].

Depending on the compound/extract to be entrapped or included, liposomes are prepared differently. For example, for the inclusion of plant extracts, volatile oils, etc. it is considered that obtaining multilayered and giant (with size up to 500 nm) liposomes is favorable [22,61]. Thus, the main method of obtaining this type of liposomes is the method that uses organic solvents for the formation of a lipid layer to be hydrated, the method called the lipid film hydration method. Hydration according to the authors Jahanfar et al., 2021 [62] is important to be done with hydrophilic compounds that give the electrical charge to the formed system (for example the phosphate buffer) to increase the stability of the emulsion.

Conventional liposome formulation methods according to Mozafari et al., 2010 [21] include four important stages: the formulation of the lipid phase to be hydrated, its hydration, purification and analysis of the formulated vesicles.

The effect of black mulberry fruit extract encapsulated in spray-dried liposomal powders was studied compared to those coated with chitosan, both included in chocolate, at different pH and temperatures (40, 60 and 80 °C). The results of the study showed that liposomal powders coated with chitosan provided better protection of anthocyanin content, both at increasing temperature and pH. In addition, encapsulation of mulberry fruit in liposomes improved the in vitro bioaccessibility of anthocyanins [9].

3.3.2. Characterization of the Size and the Electric Surface Potential of the Liposomes by DLS

The size distribution of the liposomes and the electrical surface charge (Zeta Potential) of the liposomes were further determined by DLS analysis. The analysis of these param-

eters for liposomes is of major importance because it can provides information on their stability [63–65].

The size distribution histograms of empty liposomes (L) and mulberry–loaded liposomes (MnL) are shown in Figure S2A and Figure S2B respectively.

According to Figure S2A, the sizes of L sample are between 37 and 955 nm, of which 87.12% have a size up to 500 nm, while MnL sample showed 78.68% size distribution under 500 nm (Table 3).

**Table 3.** Characterisation of liposomes.

| Liposomal Formula * | Size up to 500 nm (%) | Zeta Potential (mV) | EE% |
|---|---|---|---|
| L | 87.12 | −2.80 | - |
| MnL | 78.68 | −2.5 | 88.25 |

* L—empty liposomes, MnL—mulberry-loaded liposomes.

Liposomes of a large size are considered to be highly effective for penetration into the cell and implicitly for transporting the hydrophilic extract included in the vesicle through the membrane of the cell. Liposomes with a size of up to 500 nm are considered to have a high biocompatibility with the cell membranes of fibroblasts [61,66,67].

The electrical surface charge of the L and MnL samples were also determined. For L and MnL samples, the electric surface charge was −2.80 mV and −2.5 mV, respectively which indicate a good stability of [24,68,69].

Our results demonstrated that the black mulberry extract did not influence physico-chemical properties of liposomes. These results are consistent with the literature that used the lipid film hydration method for the encapsulation of various medicinal substances or plant extracts [23,25].

### 3.3.3. Determination of the Entrapment Efficiency (EE%) of the Mulberry Extract in Liposomes

The entrapment efficiency of the mulberry extract in liposomes formulated by the lipid film hydration method was determined. The encapsulated efficiency revealed 88.25%of black mulberry included. Data from the literature shown different percentages of encapsulation depending on the extract/compound used. For example, Miere (Groza) et al., 2021 [24] obtained a percentage of inclusion of the extract of *Stellaria media* (L.) Vill. of 92.09%, Wichayapreechar et al., 2020 [70] obtained a percentage of 77% for the extract of *Centella asiatica*, while Liu et al., 2017 [71] included a norcantharimide derivative in liposomes at a percentage of 47.6%.

### *3.4. Cell Culture and Treatment*
### 3.4.1. Cell Viability and Proliferation Assay (MTS) of Samples

Determination of cell viability represents a key stage in the formation of cell cultures [72]. A high cell viability and a degree of confluence over 80% allow the application of the scratch method in favorable conditions [72]. The human dermal fibroblasts viability (%) according to the samples used based on trypan blue exclusion test is shown in Table 4.

The dye exclusion test is used to determine the number of viable cells. The principle of test is that live cells possess intact cell membranes that exclude trypan blue dye whereas dead cells do not [73]. Based on our results, all the samples shown no significant difference compared with CTRL.

Cellular proliferation was performed by the MTS test, using different doses of mulberry extract (between 25–200 µg/mL), as well as liposome samples with or without extract (MnL and L respectively).

**Table 4.** The effect of human dermal fibroblasts viability (%) depending on the samples used.

| Samples | % of Cell Viability |
|---|---|
| CTRL | 95.25 ± 7.31 |
| ALA_50 | 94.77 ± 5.51 |
| MnE_25 | 98.55 ± 5.90 |
| MnE_50 | 97.21 ± 7.11 |
| MnE_100 | 95.48 ± 8.24 |
| MnE_200 | 96.78 ± 7.48 |
| L | 95.15 ± 6.20 |
| MnL | 95.30 ± 6.50 |

CTRL—untreated cells; ALA_50—alanine 50 μg/mL; L—empty liposomes; MnE_25—mulberry extract 25 μg/mL; MnE_50—mulberry extract 50 μg/mL; MnE_100—mulberry extract 100 μg/mL; MnE_200—mulberry extract 200 μg/mL; MnL—liposomes with mulberry extract.

The samples tested did not exhibit proliferation inhibition on NHDF cells. Significant differences were obtained between all samples compared to the control. It should be noted that a significant difference was obtained in the case of the MnL sample whose cell viability was increased by 54.15% compared to CTRL. No statistical significances were obtained between the mulberry extracts, regardless of the concentration used (Figure 3).

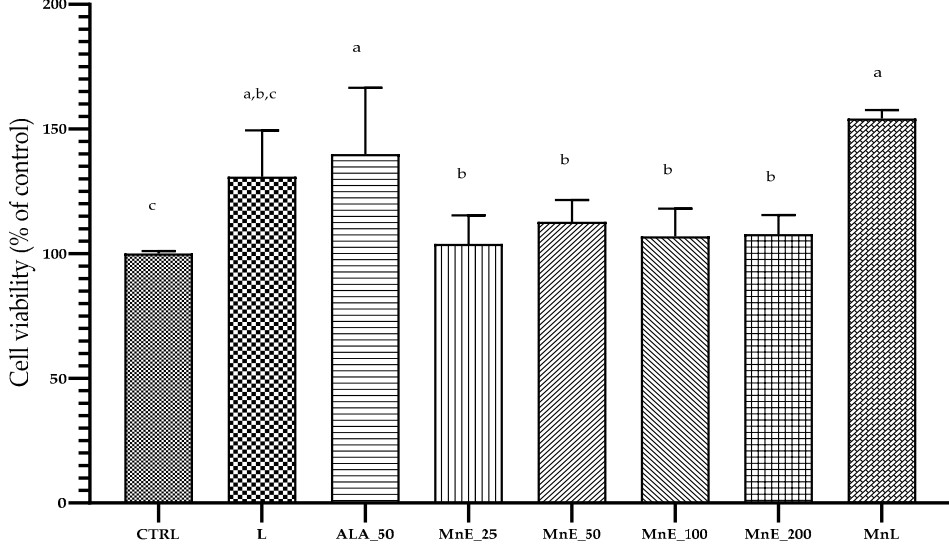

**Figure 3.** The effect of mulberry extract concentrations (25, 50, 100 and 200 μg/mL) and liposomes on cell viability of NHDF after 24 h using MTS assay. CTRL—untreated cells; L—empty liposomes; ALA_50—alanine 50 μg/mL; MnE_25—mulberry extract 25 μg/mL; MnE_50—mulberry extract 50 μg/mL; MnE_100—mulberry extract 100 μg/mL; MnE_200—mulberry extract 200 μg/mL; MnL—liposomes with mulberry extract. Different lowercase letters indicate significant differences between samples ($p < 0.05$).

The authors Im et al., 2022 [74] shown that the high content of compounds such as anthocyanins, respectively antioxidant compounds and vitamins, including vitamin C, have beneficial effects on the stimulation and proliferation of dermal fibroblasts. Hyun et al., 2021 [75] also confirm the beneficial and non-toxic effects of mulberry extracts on the process of growth, development and proliferation of fibroblasts and other dermal and hair follicle cells, demonstrating the beneficial effects on hair growth and thus combating alopecia [75].

Also, liposomal formulas composed of phospholipids and cholesterol are highly biocompatible with the human body, results that are confirmed by data from the literature [20,62,76].

3.4.2. In Vitro Testing of the Healing Effect of Mulberry Extracts and Its Liposome Using the Scratch Method

The scratch method an in vitro technique is widely used to evaluate the wound healing capacity of different plant extracts [19,40]. Wound healing is a dynamic process where activation, proliferation and migration of fibroblasts are the primary steps involved, along with other microenvironmental factors [77,78].

In the current study, normal human dermal fibroblasts (NHDFs) were treated with both different concentrations of mulberry extract (25, 50, 100 and 200 µg/mL) and mulberry extract included in liposome (25 µg/mL) for evaluation of cell migration for 0, 12, 24, 36 and 48 h (Figure 4).

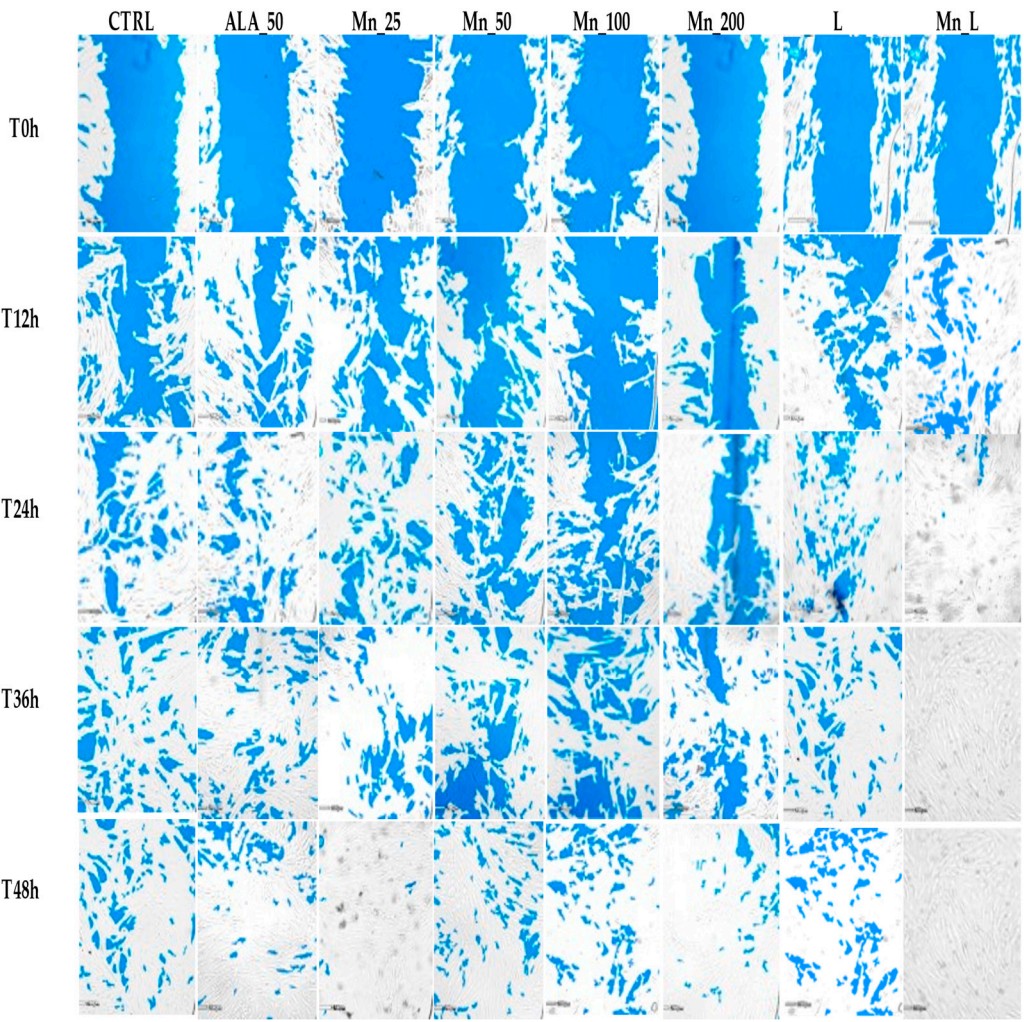

**Figure 4.** The evolution over time of the migration of normal dermal fibroblast (NHDF) for healing the wound formed in vitro by the scratch method depending on the applied treatment (different concentrations of mulberry extract (25, 50, 100 and 200 µg/mL), or liposomal formula (L and MnL) along with CTRL and positive control (Ala_50). NHDF dermal cells are represented in white and blue color shows the in vitro simulated wound that was not healed.

The images were captured using the CytoSMART Lux3BR® device. The quantitative parameters such as wound closure by area (Wnd_A) (%) were calculated using CytoSMART Lux3BR® and ImageJ software and the results are presented in Table 5.

**Table 5.** Wound closure according to the area (%), following the application of different treatments at 12, 24, 36 and 48 h.

| Samples / Hours | CTRL | ALA_50 | MnE_25 | MnE_50 | MnE_100 | MnE_200 | L | MnL |
|---|---|---|---|---|---|---|---|---|
| 12 | 35.66 ± 0.27 [a] | 47.09 ± 1.76 [a] | 50.48 ± 7.37 [a] | 43.55 ± 15.08 [a] | 30.90 ± 4.98 [a] | 46.88 ± 22.83 [a] | 44.83 ± 10.30 [a] | 40.39 ± 15.86 [a] |
| 24 | 51.40 ± 6.41 [a] | 64.19 ± 3.37 [a] | 66.70 ± 15.04 [a] | 68.33 ± 0.16 [a] | 46.62 ± 14.57 [a,b] | 72.68 ± 2.38 [a] | 73.21 ± 3.05 [b] | 81.15 ± 14.80 [a,c] |
| 36 | 66.61 ± 3.79 [a] | 74.93 ± 2.27 [a,b] | 75.87 ± 7.62 [a,b] | 70.66 ± 0.74 [a,b] | 64.78 ± 4.59 [a,e] | 78.62 ± 3.72 [b] | 94.15 ± 1.89 [c,d] | 98.78 ± 1.09 [d] |
| 48 | 82.37 ± 1.65 [a] | 92.73 ± 1.68 [b,c,d,e] | 93.81 ± 2.45 [c,d,e,f] | 84.81 ± 0.07 [a,d] | 84.63 ± 1.04 [a,d] | 89.32 ± 3.29 [d,e] | 94.20 ± 1.89 [e] | 99.72 ± 1.07 [f] |

CTRL—untreated cells; ALA_50—alanine 50 μg/mL; MnE_25—mulberry extract 25 μg/mL; MnE_50—mulberry extract 50 μg/mL; MnE_100—mulberry extract 100 μg/mL; MnE_200—mulberry extract 200 μg/mL. L-empty liposomes; MnL—liposomes with mulberry extract (25 μg/mL). Different superscripts letters indicate significant differences ($p < 0.05$) on the line (between samples at the same time (hours).

The images obtained with the CytoSMART Lux3BR® device (Figure 4) indicate the evolution over time of the closure of the "scratch" caused in vitro depending on the applied treatment.

The results showed that the most effective were the treatments with mulberry extract in a concentration of 25 μg/mL (MnE_25) and mulberry extract included in liposomes (MnL), the in vitro coverage area of the scratch was 93.81% and 99.72% respectively in 48 h.

According to the data in Table 5, it can be seen that in the case of liposomes loaded with mulberry, a statistically significant ($p < 0.05$) increase in the wound closure% is obtained starting from the time of 24 h compared to sample L. The wound healing process is an extremely complex one, involving a series of factors that act in four cascade-type stages [79,80]. The first stage is the vasoconstriction stage, namely the stage of stopping the bleeding and self-cleaning of the wound. In this stage, at the level of the wound, the transformation of prothrombin into thrombin, of fibrinogen into fibrin and subsequent formation of a crust made of platelets takes place. The next stage is the pro-inflammatory one, which is generated by the immune system, the main characteristic being vasodilatation with an increase in the concentration of oxygen and pro-healing elements at the level of the wound. Also in this stage, phagocytic cells are concentrated at the level of the wound, which practically clean the wound and act as antimicrobial elements. The MnE extract is a rich source in phenolic acids, flavonoids, phytocompounds which are proven to have antimicrobial activity and can be considered a stimulator of wound scar formation [10–12].

The proliferation phase is the third stage of wound healing. This stage begins with the phenomenon of migration of dermal fibroblasts followed by the formation of the cell matrix and re-epithelialization [81].

The process of cellular migration leads in the first phase to the covering of the wound in a monolayer of dermal cells which then end up proliferating and overlapping in several layers. An extract's composition can influence neoangiogenesis and growth factor stimulation [73–76]. Thus, we consider that the MnE extract contributed through the phytochemical composition and at this stage a different stimulation of the healinging process can be observed depending on the form of extract applied (MnEs and MnL) but also depending on the concentration applied (MnE extract 25 μg/mL having a significantly different effect compared to the rest of the applied concentrations).

In addition, according to data from literature [82–85], there is a direct relationship between the regeneration and stimulation of the third stageand the presence of compounds such as Kaemferol, derivatives of Quercetin, ferulic acid, vanillic acid and caffeic acid, compounds that were also identified in the extract of mulberry (MnE) (Table 1).

## 4. Conclusions

In this study it was demonstrated that black mulberry fruits are rich sources of bioactive compounds with high antioxidant capacity. Also, a scratch test was performed to determine the healing activity of different doses of mulberry extract. The 25 μg/mL MnE sample showed statistically significant higher healing activity compared to the control (CTRL) at 48 h.

Having the greatest cicatrizing and proliferative activity, the MnE 25 μg/mL extract was included in a liposomal formula. The encapsulation efficiency of black mulberry extract

was 88.25%, and the surface electric charge with negative values indicates good stability of the formed liposomes. Viability and proliferation assays demonstrated that mulberry extract and liposomal formulation had no adverse effects on NHDF cells. In addition, liposomes with included mulberry extract were shown to have improved simulated wound closure compared to MnE samples reaching a density of 99.72% in 48 h. A higher percentage of in vitro wound healing was observed with the application of the liposomal formulation than with the application of MnEs extracts. Further research is needed to observe the stability of the liposomes over time and to elucidate the mechanism by which both the mulberry extracts and the liposomal formulation contribute to wound closure. The results obtained in this study open new areas of research regarding the identification of the compounds from mulberry fruits, which contribute to the fibroblast proliferation process, to their synergistic or additional action. As future perspectives, we proposed to expand the research by including the liposomal formula in preparations for topical application, with the aim of using it in dermal lesions as an alternative, non-toxic and natural treatment.

**Supplementary Materials:** The following supporting information can be downloaded at: https://www.mdpi.com/article/10.3390/app13021041/s1, Figure S1: HPLC-DAD-MS (ESI+) chromatograms of the phenolic profile for black mulberry fruit at two wavelengths (280 nm, 340 nm). Figure S2. A—Histograms of the diameter distribution for L; B—Histogram of the diameter distribution for MnL. Table S1. Chemical structures of tentative compounds identified in black mulberry fruit by HPLC-DAD-MS (ESI$^+$).

**Author Contributions:** Conceptualization, S.I.V., A.R.M. and F.M.; methodology, S.I.V. and F.M.; software, A.R.M.; validation, L.V., M.G. and A.A.; formal analysis, A.R.M.; investigation, V.L. and C.P.; resources, V.L.; writing—original draft preparation, A.R.M. and F.M.; writing—review and editing, S.I.V., A.R.M. and F.M.; supervision, S.I.V.; project administration, L.V. All authors have read and agreed to the published version of the manuscript.

**Funding:** This research received no external funding.

**Institutional Review Board Statement:** Not applicable.

**Data Availability Statement:** Not applicable.

**Acknowledgments:** Adriana Memete acknowledges the support provided by CytoSMART Technologies B.V. Eindhoven, Netherlands for the CytoSmart Lux3BR® microscope, won through the "CytoSMART Research Grant 2021" grant competition. The authors acknowledge the support provided by the University of Oradea through the grant competition "Scientific Research of Excellence Related to Priority Areas with Capitalization through Technology Transfer: INO—TRANSFER—UO", Project no. 309/2021 and Project No. 246/2022.

**Conflicts of Interest:** The authors declare no conflict of interest.

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
