# Peer review of "An In Vitro Study of the Healing Potential of Black Mulberry (Morus nigra L.) Extract in a Liposomal Formulation"

_applsci, doi:10.3390/app13021041_

Round 1
Reviewer 1 Report
The paper entitled: "An in vitro study of the healing potential of black mulberry (Morus nigra L.) extract in a liposomal formulation", submitted by Ramona Memete et al., is an important paper that discusses for the first time the cytotoxic and healing efficacy of black mulberry, in both forms, as an extract and encapsulated in a liposome, on normal human dermal fibroblasts (NHDF). the phytochemical profile and the antioxidant activities were also investigated in this study. This paper is well-written, however, some queries must be addressed, before considering the paper.
* Please add a reference to support the methodology used for cell viability testing with trypan blue exclusion assay.
* Table 1, HPLC-DAD-MS (ESI+), rectify the name of molecules. e.i. Kaempferol 3-O-glucuronide; Cyanidin-3-O-glucoside; Cyanidin-3-O-rutinoside ... see PubChem for the exact spelling of each molecule.
* Table 1: RT is unnecessary because no separation was performed. On what basis abundance was assessed?
* Alkaloids are usually analyzed in positive mode (ESI+), why the authors used HPLC-DAD-MS in a positive electrospray ionization? (The ionization mode in MS is unusual - alkaloids more easily accept protons and therefore, they are commonly analyzed in a positive mode.)
* Usually the obtained results (mass spectra) should be compared with literature or appropriate databases. add to Table 1, the corresponding information.
* No standards are mentioned in Materials and Methods section, for the phytochemical analysis.
* Add the chemical structures along with their CID (Compound ID, either from PubChem or Zinc or any relevant database).
* Line 275, the equation 2, please rectify, the number of living cells.
* Because of the unique properties of liposome technology, researchers have been working tirelessly on the successful creation of innovative drug delivery vehicles based on liposomes. However, there are certain limits to employing liposomes for biomedical applications because to their low stability, which is mostly the reason of fast drug leakage inside the aforementioned matrices. The authors in this study did not discuss these limitations in their study.
Author Response
Our responses to reviewer 1 comments are uploaded as a Word file. Please see the attachment.

Reviewer 2 Report
The work entitled “An in vitro study of the healing potential of black mulberry (Morus nigra L.) extract in a liposomal formulation” reports the characterization of black mulberry (Morus nigra L.) extract and its encapsulation for the effective treatment of dermal damage. The work is interesting; however, some improvements need to be done before acceptance.
1. The abstract should be a total of about 200 words maximum. Too many details from the result section have been included. Please reduce the abstract length.
2. In the introduction it is highlighted that the functions of the dermal layer are better supported by the nutrients supplied by the blood stream, particularly for vitamin C. But then in the conclusion the authors propose to include the liposomal formula in preparations for topical application.
3. The authors mention that one of the advantages of using liposomal systems at the dermis level is that drug delivery occurs in a controlled and continuous manner, allowing for a more reproducible treatment. But this effect highly depends on the composition of the liposomal formulation, and non-triggerable liposomes still lack reproducibility in the clinic. Please clarify.
4. The authors did not perform release studies under different conditions (temperature, buffer, pH, etc). More references of liposomes with the same composition as the used in this manuscript should be included or the authors should consider avoiding the term “controllable release” to characterize their formulation.
5. If previously reported by some of the authors that liposomes composed of PS are more stable based on size and charge, why use PC in this case?
6. From DLS measurements, indicate average size of liposomes and distribution (not only distribution). Consider presenting a table with average size, distributions and SD.
7. Explain why the size of the liposomes was decided to be “giant” instead of the more usual nano.
8. Line 67: Rephrase: fibroblasts are precursors of dermal collagen.
9. Figure 1: Abbreviations have not been defined. Reorganize (not all concentrations were encapsulated, confusing-too many arrows-).
10. Line 216: …described by…
11. Line 245: defined if MnEs is the plural for MnE.
12. Section 2.8 should only describe general cell culture. The scratch is described again in subsection 2.8.3. Please eliminate lines 252-262.
13. The MTS Assay is a Cell Proliferation Assay. Please substitute all the “cytotoxicity” for “proliferation” throughout the manuscript.
14. Alanine is defined as the positive control that would promote cell proliferation and wound healing. The liposomal formulation, although used to assess the effect its effect on the fibroblasts and determine if it has a detrimental effect because it is part of the encapsulation, should not, on its own, promote cell proliferation, so it cannot be defined as a positive control.
15. If the scratch assay was monitored for 12-48h, why the trypan blue and the MTS assay were performed after 24h of incubation with the test samples?
16. Line 370: Correct “and the phenolic acids and flavonoids would could prove important”.
17. Line 406 states that more than one liposomal formulation was evaluated.
18. Line 460: Reference#57 does not relate with the text.
19. Line 470: size distribution does not have an effect on the in vitro results?
20. Line 485: is negative charge sufficient to state that the liposomal formulations are stable?
21. Line 487: reformulate, “the percentages of liposomes up to 500 nm being close.”
22. Table 4: clarify what samples (significant differences) are the superscripts letters comparing.
23. Please comment on why the % of closure is greater for the liposomal formulation compared to the positive control (ALA_50) at all the times tested.
24. Explain why the effect is higher when the extract is encapsulated rather than when is free and available to the cells.
25. Line 612: what control (media only, alanine, empty liposomes).
26. Line 623: various factors, such as…
Author Response
Our responses to reviewer 2 comments are uploaded as a Word file. Please see the attachment.

Round 2
Reviewer 1 Report
As the authors have included all the suggested revisions, the paper now is suitable for publication in Applied Sciences.